# Self-Assembling Graphene Layers for Electrochemical Sensors Printed in a Single Screen-Printing Process

**DOI:** 10.3390/s22228836

**Published:** 2022-11-15

**Authors:** Andrzej Pepłowski, Filip Budny, Marta Jarczewska, Sandra Lepak-Kuc, Łucja Dybowska-Sarapuk, Dominik Baraniecki, Piotr Walter, Elżbieta Malinowska, Małgorzata Jakubowska

**Affiliations:** 1Printed Electronics, Textronics & Assembly Lab, Center for Advanced Materials and Technologies CEZAMAT, Warsaw University of Technology, 19 Poleczki, 02-822 Warsaw, Poland; 2Institute of Metrology and Biomedical Engineering, Warsaw University of Technology, 8 A. Boboli, 02-525 Warsaw, Poland; 3The Chair of Medical Biotechnology, Faculty of Chemistry, Warsaw University of Technology, 3 Noakowskiego, 00-664 Warsaw, Poland; 4Division of Medical Diagnostics, Center for Advanced Materials and Technologies CEZAMAT, Warsaw University of Technology, 19 Poleczki, 02-822 Warsaw, Poland

**Keywords:** printed electronics, colloid suspensions, shear flow, shear thinning, electrochemically active surface area, voltammetric electrodes, self-assembling material

## Abstract

This article reports findings on screen-printed electrodes employed in microfluidic diagnostic devices. The research described includes developing a series of graphene- and other carbon form-based printing pastes compared to their rheological parameters, such as viscosity in static and shear-thinning conditions, yield stress, and shear rate required for thinning. In addition, the morphology, electrical conductivity, and electrochemical properties of the electrodes, printed with the examined pastes, were investigated. Correlation analysis was performed between all measured parameters for six electrode materials, yielding highly significant (*p*-value between 0.002 and 0.017) correlations between electron transfer resistance (Ret), redox peak separation, and static viscosity and thinning shear-rate threshold. The observed more electrochemically accessible surface was explained according to the fluid mechanics of heterophase suspensions. Under changing shear stress, the agglomeration enhanced by the graphene nanoplatelets’ interparticle affinity led to phase separation. Less viscous pastes were thinned to a lesser degree, allowing non-permanent clusters to de-agglomerate. Thus, the breaking of temporary agglomerates yielded an unblocked electrode surface. Since the mechanism of phase ordering through agglomeration and de-agglomeration is affected by the pastes’ rheology and stress during the printing process and requires no further treatment, it can be appropriately labeled as a self-assembling electrode material.

## 1. Introduction

Screen-printed electrodes (SPEs) have been considered one of the most valued tools for electrochemical analysis for decades [1,2,3,4,5,6,7,8,9]. This is mainly due to SPEs’ low-cost fabrication technology that is easily scalable from single-piece laboratory manufacturing to large-scale mass production. Moreover, virtually all material types can be incorporated into screen-printing pastes [10,11,12,13,14,15,16].

Given the high demand for more effective SPEs, research towards understanding, explaining, and optimizing the printing and pastes’ fabrication technology would seem natural. However, there is little to be found in scientific literature focusing on a direct comparison of screen-printing pastes technology and SPEs’ analytical performance. Three notable exceptions are the works of Jewell et al. [17], Somalu et al. [18], and Chiticaru et al. [19]. The authors focus on the SPEs’ fabrication efficiency in the first study. However, regardless of significant references to the sensor applications, the parameters analyzed in the article include only rheological and electrical ones. The second publication concerns SPEs modified by the drop-casted suspension of graphene oxide in different concentrations. The third work describes the effect of the binder concentration in the printing pastes on the electrochemical impedance and electron transfer resistance of the SPEs. There are, however, almost no other more elaborate studies. Most of the literature could be classified as either case reports of, e.g., new material’s successful employment [11,20,21,22,23] or applied research focusing on the analytical performance and material of SPEs being commercially available [24,25].

On the other hand, extensive research is available regarding rheology and its relationships to both the contents of the pastes and the properties of printed devices. From the mechanical models of paste flow through printing mesh [26], high-speed video imaging of the printing process [27], to the effects of the rheology on the pastes’ printability and printed strain sensors’ performance [28], to the influence of the carbon black/graphite ratios on the pastes’ viscosity and printed surface roughness [29].

The authors agree with Suresh et al.’s [30] claims that comprehensive work should be carried out to fill the literature deficiency in this aspect. In the vein of applying rheological and composite theories, the influence of the RuO_2_ content on the performance of the potentiometric sensors was reported by Pepłowski et al. [31]. By applying the percolation theory of composite materials [32,33,34,35,36], the observed changes in the sensitivity were explained by Pepłowski et al. Basing on their findings and recounting observations from the applied research this article was part of, the authors planned and performed the results and conclusions presented herein. This work should be treated as a preliminary study regardless of the multi-variable approach of investigating relationships between numerous parameters. Its main goal was to extract meaningful information from experience accrued over previous works and formulate a direction for future studies. A more accurate model could then be established with a narrowed scope of included parameters and more rigorous statistical analysis.

The work described herein comprised preparing a batch of experimental pastes with varying carbon fillers and total filler content, measurements of the pastes’ rheology, voltammetric response, electron transfer resistance, and electrochemically active area surface as well as electrode layer thickness, electrical resistivity, and surface roughness. Then, correlation analysis was performed, and significantly related parameters were selected based on the results. Conclusions were drawn from the observed correlations supported by the available heterophase fluids and suspension models. Finally, the outline of the following experiments was proposed to verify the conclusions of this work.

## 2. Materials and Methods

### 2.1. Materials and Reagents

Potassium hexacyanoferrate (III) (K_3_Fe(CN)_6_), Potassium hexacyanoferrate (II) (K_4_Fe(CN)_6_) were purchased from Sigma Aldrich (Schnelldorf, Germany). Potassium chloride and ethanol were purchased from Avantor Performance Materials Poland S.A., Poland. Polymethyl methacrylate (PMMA), MW ≈ 350′000, butyl diglycol acetate (OKB) for dissolving polymer matrix, and dibutyl phthalate (DBP) were acquired from Merck Polska (Warszawa, Poland). Graphene nanoplatelets (GNPs) with a mean diameter of 25 µm and thickness below 8 nm were supplied by XG Sciences Inc. (USA) under the trade name GNP-M25. Polyester (PET) foil MYLAR^®^ with a thickness of 125 µm was fabricated by DuPont Teijin Films (Dumfries, UK) and delivered by Selmex (Plewiska, Poland).

Acting as a benchmark reference for the experimental pastes, a commercially available PF-407A screen-printing paste was purchased from Henkel Polska sp. z o.o.

### 2.2. Preparation of Printing Pastes

All experimental pastes’ compositions are listed in Table 1. The vehicle for them was an 8 wt% PMMA solution in OKB. PMMA granulate was dissolved in OKB over 48 h with constant stirring and heating to 40 °C using a magnetic stirrer. To the PMMA vehicle, the functional phase was added, as specified below.

In all the experimental pastes, the primary filler material was GNPs. The respective amounts were added to the vehicle and mixed thoroughly in an agate mortar until a homogenous composition was obtained. Then, to break down the GNPs agglomerates, the paste was rolled twice in a three-roll-mill Exact 80E with SiC rolls, a gap between rolls set to 5 µm, and a feed roller speed of 100 rpm. After rolling, in the case of the pastes containing CB or DBP, the respective material was added in the amount specified in Table 1 and mixed thoroughly in an agate mortar. The compositions containing CB were not mixed in a three-roll mill to prevent the breaking down of CB particles.

### 2.3. Fabrication of the Electrodes

All the electrodes were fabricated using 77 T (77 threads/inch) polyester screens masked with a UV-curable pattern supplied by Sico Polska sp. z o.o. (Warszawa, Poland). The pattern consisted solely of carbon-based composite employed as electrode materials and electrical contacts for the connection with the measurement setup (Figure 1).

As a substrate for printing, 125 µm thick PET foil was used. The foil sheets (140 × 210 mm) were pre-heated for 40 min at 130 °C to prevent the influence of material shrinkage on the SPEs’ geometric surface area during the thermal curing process. The printing was performed using an Aurel C920 (AUREL s.p.a., Modigliana, Italy) semi-automatic printer. The printing process parameters (squeegees’ low position, deflection and movement speed, screen position) were adjusted for each paste to obtain optimal printability, i.e., printing a continuous layer in the first squeegee pass. After the carbon paste layer was deposited, the electrodes were cured at 120 °C for 20 min and then vacuum-sealed until the measurements were carried out. In the case of PF-407A paste, it was employed for printing in two batches. The paste was applied on the printing screen as supplied in the sealed container for the first one. For the second batch, the container with the paste was stirred in a planetary centrifugal mixer Kakuhunter SK-350T II (Honeystone Ltd., London, UK), with a revolution rate of 2200 rpm and a rotation speed of 70 rpm.

### 2.4. Physical Measurements

The rheology of the pastes was measured using a rotational rheometer BROOKFIELD RS CPS+. Electrical resistance was measured by a UNI-T ohmmeter UT 804 in a two-point setup. The morphology of the printed layers was measured using a DektakXT stylus profilometer (Bruker Corp., Billerica, MA, USA). Scanning electron microscopy (SEM) was conducted on a Hitachi SU8230 (Hitachi High-Tech Europe GmbH, Krefeld, Germany) instrument with an accelerating voltage of 5.0 kV and an upper secondary electron detector.

### 2.5. Electrochemical Measurements

Cyclic voltammetry (CV), square-wave voltammetry (SWV), and impedance spectroscopy (EIS) were performed using CHI 660A, 660D, and 1040A electrochemical workstations (CH Instruments, Bee Cave, TX, USA). The experiments were executed using a three-electrode system containing a working electrode printed with various carbon-based pastes, an Ag/AgCl/1.0 molL-1 KCL reference electrode (Mineral, Łomianki, Poland), and a gold wire that served as the auxiliary electrode (Sigma Aldrich, Darmstadt, Germany). All the electrochemical studies were performed at room temperature. Cyclic voltammetry was conducted with the potential window from -0.6 to 1 V at scan rates from 5 to 0.05 Vs^−1^, or 0.1 Vs^−1^. Square-wave voltammetry was executed at a pulse amplitude of 15 mV, an increment of 4 mV, and a frequency of 15 Hz. EIS measurements were performed at a DC potential of 0.2 V and an amplitude of 0.005 V in the frequency range from 1 to 100 kHz. The electrochemical measurements were conducted using a redox indicator −5 mM ferri/ferrocyanide in 0.1 M KCl solution.

### 2.6. Statistical Analysis

The correlation analysis of the parameters obtained was performed using MATLAB (release R2021b, The MathWorks, Inc., Natick, MA, USA). The function implemented for calculating the correlation coefficient and the respective *p*-values was corrcoef. The syntax for the function corrcoef, as provided by The MathWorks, Inc. states that:

“[R,P] = corrcoef(___) returns the matrix of correlation coefficients (R) and the matrix of *p*-values (P) for testing the hypothesis that there is no relationship between the observed phenomena (null hypothesis). If an off-diagonal element of P is smaller than the significance level (α, default is 0.05), then the corresponding correlation in R is considered significant.” The null hypothesis in the case considered herein was: H_0_: “There is no linear relationship between the analyzed parameters.” In this case, the value α ≤ 0.05 indicates that the probability of stating that there is a significant relationship between parameters was not greater than 5%. The calculated *p*-values represent the probabilities of obtaining the analyzed parameters’ values if they were randomly (independently) distributed, i.e., the *p*-value expresses the probability of basing the test on coincidental data.

## 3. Results and Discussion

All measured values were collected in Table 2. The cyclic voltammetry studies performed in the presence of a 5 mM ferri/ferrocyanide redox indicator showed that the lowest potential difference between anodic and cathodic peak currents was obtained for electrodes formed of GNP-4 paste reaching 0.43 V, followed by GNP-3, for which peak current separation was of 0.62 V (see Figure 2). This indicates that the electrode surface was blocked to a lower extent than electrodes GNP-1, GNP-2, and PF-407 A. Such results were further confirmed with the use of impedance spectroscopy (Figure 3), where the lowest charge transfer resistance (R_et_ = 3.12 kΩ) was reached for the working electrode made with the GNP-4 paste.

The lesser the pastes’ viscosity (η) at the shear rate (γ˙ ≈ 0), the lesser the electron transfer resistance (Ret). The higher the shear rate (γ˙) at which the pastes’ viscosity (η) dropped to ≈0, the lesser was peak separation (ΔE) and cathodic peak current (ic). As described in Section 2.6., the above correlations were tested with α ≤ 0.05 (over 95% significance), as shown in Table 3. From that, it may be concluded that the pastes with higher initial viscosity and thinning occurring at a higher shear rate are more likely to yield electrodes with an active surface area blocked to a lesser degree. The explanation for these observed relationships is discussed in the following paragraphs and was based on fluid mechanics.

Before entering that discussion, it is worth noticing that the PF-407A paste exhibited highly thixotropic properties. Mixing influenced its rheology and resulting electrical and electrochemical characteristics to a great degree. Most importantly, the paste’s viscosity increased after mixing, which may seem counterintuitive. At the same time, the printed layer’s electrical conductivity of the mixed PF paste is greater than that of the other materials by a half level of magnitude. The reasons for that should become apparent after laying out the primary explanation for the observed correlations.

### 3.1. General Rheological Properties of Printing Pastes

Printing pastes are colloid suspensions, with the continuous medium of liquid polymer solution and the dispersed phase commonly called a filler in the field of printing technology. Shear-thinning behavior is a general quality required for the colloidal suspensions to be applicable in printing [26,37,38,39,40,41]. Due to the initial high viscosity, the paste remains stable on the screen before squeezing. Subsequently, due to the shear stress applied through the squeegee’s motion, the paste *thins*, i.e., its viscosity drops close to 0. In terms of the quality of the print, too low viscosity is undesirable. The material in such a state pours over the substrate without forming the pattern applied on the screen. Thus, in the range of shear rate below the squeegee motion (i.e., ≈180 s^−1^) [37,38], the paste should exhibit non-negligible viscosity. Only GNP-2 displayed nonsufficient viscosity (Figure 4), as it may be inferred from the surface roughness (R_a_) that was substantially higher than in the case of the other materials. It is essential to note in this regard that this paste had the lowest carbon content, which contrasts with the highest R_a_ value.

### 3.2. Influence of Shear-Thinning on the Filler Particles and Resulting Surface Morphology

It was demonstrated by the theoretical models of colloidal suspensions that a shear flow may result in increased interaction between suspended particles [42,43]. Thus, the particles may exhibit a higher tendency for agglomeration than in a static state. On the other hand, GNPs, in general, display a significant affinity for creating interparticle bonds. Given these two factors, intensive re-agglomeration of GNPs dispersed in the paste is understandable.

What the above enables—and is not present in the literature on the materials for SPEs—is that rapid agglomeration of the particles dispersed in the colloid causes phase separation [43]. The less viscous liquid phase thus flows to the bottom layer while the agglomerated particles emerge on the top layer. Given the metastable characteristic of such shear-induced agglomerates [44], the higher electrochemically active surface area values and lesser electrode blocking can be explained.

The carbon filler particles, especially GNPs, which exhibit a high affinity for inter-facial attraction, agglomerate under the shear applied during the printing process. The polymer solution flows in the bottom layer, and the carbon material dominates in the surface layer. After the shear declines, i.e., when the paste is deposited on the substrate, the metastable agglomerates decay, exposing the particles’ facets uncovered by the polymer. For this mechanism to occur, several conditions have to be fulfilled:The suspended particles’ concentration must be sufficient for the interparticle attraction to offset repulsive forces [44];The particles’ dimensions have to be small enough [42];Liquid phase’s shear-thinning is needed for phase separation [43].

At the same time, the non-spherical shape of the particles, contributing to more pronounced interparticle attraction, enhances the described process. In addition to exposing the unblocked electrode surface, the paste should conform to the printed pattern, which requires non-negligible viscosity after flowing through the screen, as explained in Section 3.1. The above considerations were confirmed by scanning electron microscopy (SEM).

The previously noted fact that the GNP-4 paste exhibited the highest asymptotic viscosity (Table 2) should also be commented on in light of the presented explanation. The agglomerates of GNPs separate as a phase from the polymer solution when they are formed under screen-printing shear. Thus, two different flows could be distinguished: the GNPs and the polymer vehicle. The latter is expected to show minimum viscosity since the polymer chains should align with the flow direction [44,45]. On the other hand, the former GNPs’ flow mainly includes solid particles with strong interparticle attraction forces. Hence, that flow exhibits much higher viscosity. The result of the simultaneous flux of these two different flows is a higher viscosity even at the shear rates thinning other pastes to η values much closer to 0 Pa·s.

Attaining that property, i.e., the non-negligible asymptotic viscosity, could be labeled as the indicator of the phase-separation flow occurrence. At the same time, the prediction variables for that are the static viscosity, which should be as low as possible, and the high shear rate required for the thinning to occur. That points toward the final conclusion about the observed mechanism. The only difference between the GNP-3 and GNP-4 pastes is the addition of dibutyl phthalate (DBP). That addition was made based on the authors’ previous technological experiences in the field of printed electronics. The DBP amount was adjusted manually for the apparent change in the paste’s viscosity. The process of adjusting the DBP content is not reported herein.

Nevertheless, the DBP’s influence on the rheology is thinning the polymer solution to even lower viscosity values at the static condition. That directly influences the first listed prediction variable, i.e., static viscosity (η_0_). However, it also relates to the shear-rate threshold for thinning since the less viscous polymer vehicle enables a more pronounced ordering of the paste’s phases. Moreover, the process of phase-ordering leads to a higher viscosity at higher shear rates, as explained in the previous paragraph. Thus, the added thinning solvent (DBP) led to the transition from the GNP-3 electrodes with a more blocked active surface area to the less blocked one achieved using the GNP-4 paste.

Here, it should also be explained why the seemingly counterintuitive changes in the PF paste properties occurred, as stated earlier. The increase in viscosity after mixing was observed with the simultaneous rise of the electrical conductivity. That might be ascribed to the homogenizing of the paste through high-revolution mixing. The carbon filler particles were distributed more uniformly in the polymer vehicle, thus resulting in much higher conductivity. At the same time, the dispersed, non-agglomerating particles do not undergo the mechanism observed mainly for the GNP-4 paste, i.e., the PF paste was based on more spherical and less-aggregating material (probably carbon black and graphite instead of GNPs, as might be inferred from the SEM images, Figure 5), and does not separate into a two-phase flow. Thus, the paste’s viscosity drops significantly at the higher shear rates while assuming higher values in the static conditions.

## 4. Conclusions

Although the above explanation for the observed results is well-established in the available literature on the rheology of colloid suspensions, it should be noted that little to no research was conducted specifically in the field of electrochemistry of printed layers. The referenced works about rheology and phase ordering in the shear flow are concerned primarily with mining and metallurgy or organic matter. Further studies must be undertaken to bridge this significant gap and allow more controlled development of the SPEs technology.

The presented approach allowed for a relatively unblocked, electrochemically active surface area of the SPEs with only printing and curing processes. Future studies should be aimed at the adjustment of the content of the paste used for the elaboration of electrodes that would allow for further lowering of the current peak separation as well as lower charge transfer resistance. The possibility of attaining ready-to-use electrodes in just one technological stage without a need for further modification is undoubtedly a very valuable opportunity in the field of acclaimed analytical tools such as printed sensors. That can be accomplished by employing the self-assembling properties of the graphene nanoplatelets as a filler phase in printing pastes. To reach that possibility, further research should encompass a detailed investigation of the influence of paste-thinning solvents (such as DBP employed herein) and a more exhaustive statistical analysis of correlations between observed variables. That will lead to a more precise definition of the theoretical flow model that was only broadly outlined in this article. Finally, the analysis should allow for accurate experimental hypotheses about applying the proposed model to fine-tune pastes’ properties for efficient SPEs’ fabrication. Testing such hypotheses is required to prove the assumed approach for employment in the mass-scale production processes.

## Figures and Tables

**Figure 1 sensors-22-08836-f001:**
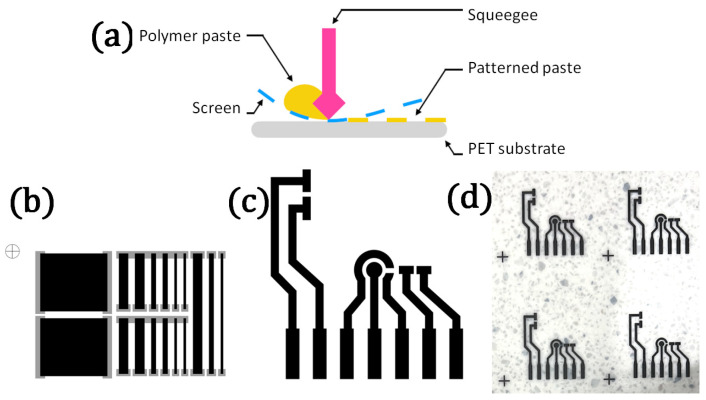
(**a**) Schematic of the screen-printing method; (**b**,**c**) printing patterns for testing sheet resistance (**b**), gray—silver contacts and sensor electrodes (**c**); (**d**) PET sheet with printed electrodes.

**Figure 2 sensors-22-08836-f002:**
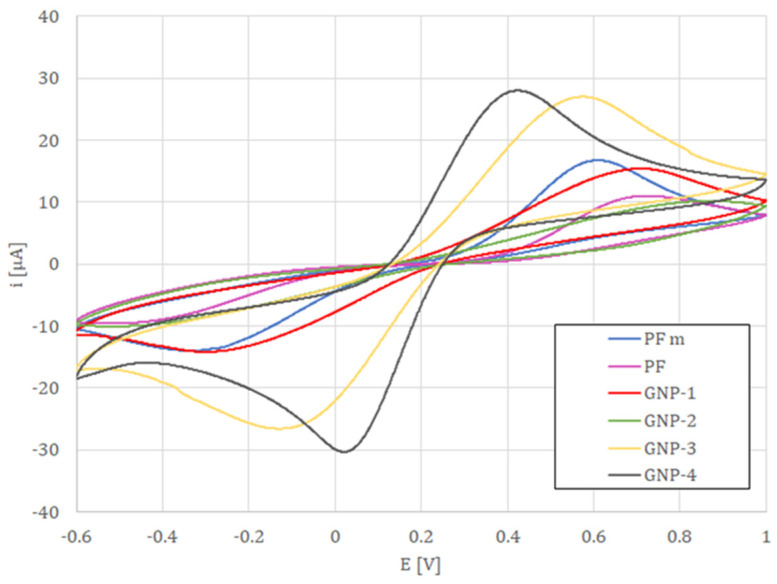
CV measurements of SPEs fabricated with the examined pastes vs. Ag/AgCl/1.0 mol L^−1^ KCL reference electrode and a gold wire as the auxiliary electrode in 5 mM ferri/ferrocyanide in 0.1 M KCl solution; scan rate of 0.1 Vs^−1^.

**Figure 3 sensors-22-08836-f003:**
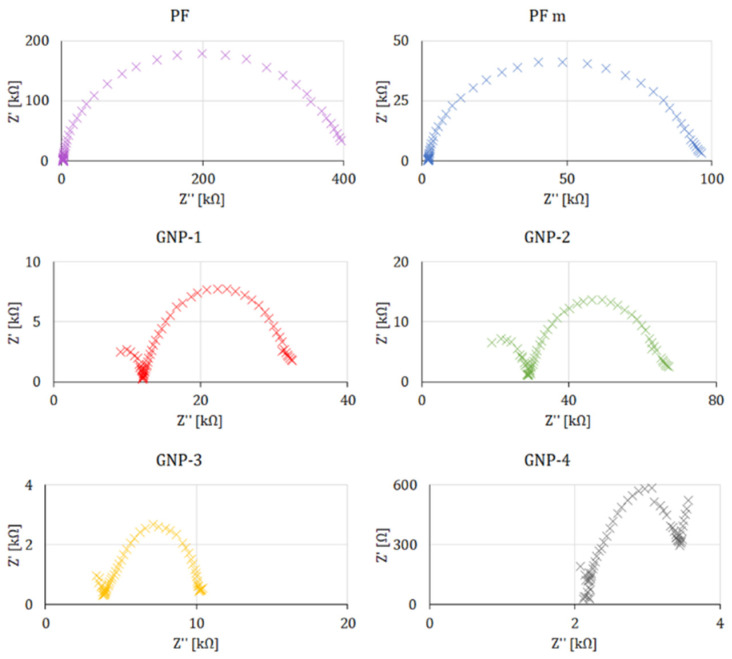
EIS measurements of the examined electrodes at a DC potential of 0.2 V, the amplitude of 0.005 V in the frequency range from 1 to 100 kHz vs. Ag/AgCl/1.0 mol L^−1^ KCL reference electrode, and a gold wire as the auxiliary electrode in 5 mM ferri/ferrocyanide.

**Figure 4 sensors-22-08836-f004:**
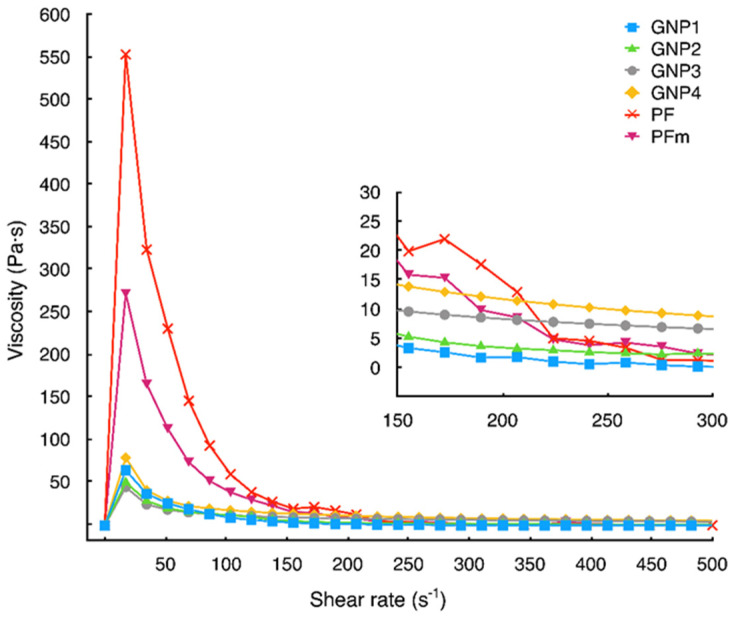
Viscosity curves for the examined pastes; inset—magnification of the curves in the shear-rate range corresponding to pastes squeezing through the printing screen.

**Figure 5 sensors-22-08836-f005:**
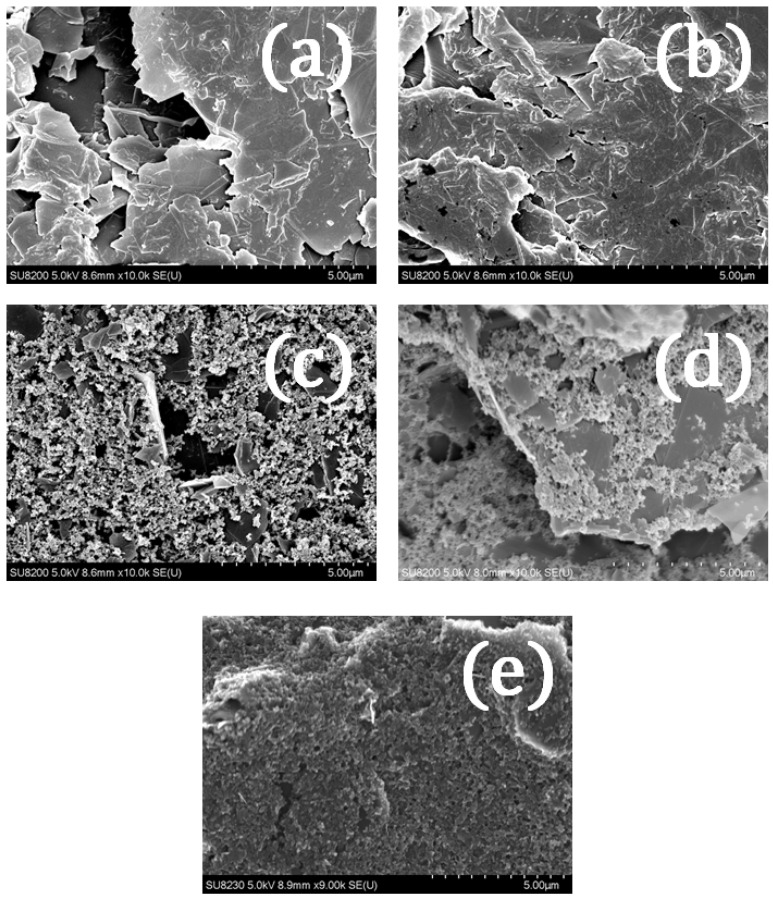
SEM images of SPEs carbon layer; (**a**,**b**) GNP-1 and GNP-2 layers, respectively, with clearly visible GNPs’ surfaces that are covered by a bulging polymer layer; (**c**) GNP-3 with visible addition of CB to the layer but covering the GNP’s surface; (**d**) GNP-4, covered by a polymer matrix to a much less extent; instead, carbon black particles are visible covering the GNPs. (**e**) PF-407 being covered by the polymer (probably less viscous than in experimental pastes GNP-1 through -4).

**Table 1 sensors-22-08836-t001:** Compositions of experimental pastes prepared for the study.

Paste Symbol	GNP Content (wt%)	CB Content (wt%)	DBP Content (μL/g)
GNP-1	15	0	0
GNP-2	12	0	0
GNP-3	10	3	0
GNP-4	10	3	100

**Table 2 sensors-22-08836-t002:** Physical and electrochemical properties of the layers screen-printed with the examined pastes.

	Material	GNP-1	GNP-2	GNP-3	GNP-4	PF-407 A w/Mixing	PF-407 A w/o Mixing
Parameter							
σ electrical conductivity	S	513.6	612.7	1248.56	1027.99	1684.24	530.33
γ˙_th_ thinning shear rate for η ≈ 0	s^−1^	500.0	327.6	999.9	1000.0	500.0	500.0
η_0_ static viscosity at γ˙ = 0	Pa·s	50.9	65.0	66.9	79.9	554.5	273.3
η_p_ viscosity at γ˙ = 172 s^−1^	Pa·s	4.3	2.6	9	12.9	15.3	21.9
η_∞_ asymptotic viscosity, γ˙→∞	Pa·s	1.0	0.1	1.3	3.1	0.6	0.3
γ yield stress	Pa	1200	1800	900	1600	15,000	8000
R_a_ surface roughness	μm	5.2	7.1	5.6	6.5	2.2	2.1
R_et_ electron transfer	kΩ	42.4	21.7	8.37	3.12	88.7	163.0
ΔE_redox_ peaks potential separation	V	1.24	1.03	0.62	0.43	0.99	1.01
i_a_ anodic peak current	μA	0.525	1.57	4.04	5.54	4.35	4.52
i_c_ cathodic peak current	μA	−0.409	−1.08	−6.18	−10.3	−2.22	−2.12
electrochem. surface	mm^2^	0.36	0.39	1.67	1.43	1.03	1.46

**Table 3 sensors-22-08836-t003:** Statistically significant correlations between measured parameters.

Parameters Correlated	Pearson’s Correlation Coefficient ρ	*p*-Value
R_et_ ↔ η_0_	96.3%	0.0020
ΔE_redox_ ↔ γ˙_th_	−89.2%	0.0168
i_c_ ↔ γ˙_th_	−91.0%	0.0118

## Data Availability

Not applicable.

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
