# Peer review of "Self-Assembling Graphene Layers for Electrochemical Sensors Printed in a Single Screen-Printing Process"

_sensors, 2022, doi:10.3390/s22228836_

Round 1

Reviewer 1 Report

This manuscript reported experiments of pastes and printing parameters of screen-printing process for SPE. Although the novelty is weak, the experimental results are useful for other researchers. 

1. Figures of the screen-printed electrodes should be presented.

2. Providing a schematic of the fabrication process of the  screen-printed electrode will be better.

3. In Table 2, the electrode surface was different between electrodes, can the author give an explain of that.

4. Please provide SEM image of SPE prepared with other pastes, GNP-1, GNP-2, GNP-3.

Reviewer 2 Report

Review of the manuscript Self-assembling graphene layers for electrochemical sensors printed in a single screen-printing process».

The considered manuscript is devoted to the analysis of correlations between electrochemical and rheological parameters of electrodes of electrochemical sensors with graphene content. The results of the analysis are interesting and may be useful for researchers. However, to improve the quality of the manuscript, it is necessary to disclose in more detail the methodology for finding correlations and correct a number of typos and inaccuracies.

The main remark.

It is necessary to reveal in more detail the methodology of finding correlations. It is necessary to provide  dependencies and equations, on the basis of which the Pearson coefficient and the value "p" are calculated in Table 3. The analysis of the values given in Table 2 does not indicate a linear relationship between the parameters.

Inaccuracies and typos.

1. L40, L61, L65, L205, L219 – typos.

2. On page 3 there are no references to Table 1, which is located at the beginning of page.4.

3. L166 What does the number in parentheses mean?

4. L169 What is the parameter α? Table 3 does not have this parameter, but there is a parameter p, which is also not characterized.

5. L283. Incorrect section numbering.

Round 2

Reviewer 2 Report

The authors of the manuscript answered the questions and corrected the inaccuracies.
However, they did not describe the parameters of correlation analysis - α and p. The authors in the response to the reviewer (but not in the manuscript) indicated that "Parameter α and p-value are statistical measures in hypotheses testing". It really is. However, in various literary sources, the parameters α and p may have other letter designations.In addition, any variable in the manuscript must be characterized in order for the reader to fully understand the text of the manuscript.

I believe that after these minor additions, the manuscript can be published.

Author Response

Thank you for clarifying the issue with statistical parameters. We will try to include a brief explanation of α and p-value to ensure full comprehensibility of the article.